# Perforator versus Non-Perforator Flap-Based Vulvoperineal Reconstruction—A Systematic Review and Meta-Analysis

**DOI:** 10.3390/cancers16122213

**Published:** 2024-06-13

**Authors:** Séverin Wendelspiess, Loraine Kouba, Julia Stoffel, Nicole Speck, Christian Appenzeller-Herzog, Brigitta Gahl, Céline Montavon, Viola Heinzelmann-Schwarz, Ana Lariu, Dirk J. Schaefer, Tarek Ismail, Elisabeth A. Kappos

**Affiliations:** 1Department of Medicine, University of Basel, 4056 Basel, Switzerland; severin.wendelspiess@unibas.ch (S.W.); dirk.schaefer@usb.ch (D.J.S.); tarek.ismail@usb.ch (T.I.); 2Department of Plastic, Reconstructive, Aesthetic and Hand Surgery, University Hospital Basel, 4031 Basel, Switzerland; loraine.kouba@usb.ch (L.K.);; 3University Medical Library, University of Basel, 4051 Basel, Switzerland; 4Surgical Outcome Research Center, University Hospital Basel, 4031 Basel, Switzerland; 5Department of Gynecology and Gynecological Oncology, University Hospital Basel, 4031 Basel, Switzerland; 6Faculty of General Medicine, University of Medicine and Pharmacy ‘Iuliu Hațieganu’, 400347 Cluj-Napoca, Romania

**Keywords:** vulvoperineal, perforator flap, non-perforator flap, cancer, reconstruction, female, surgical outcomes, quality of life, complications

## Abstract

**Simple Summary:**

Vulvoperineal defect reconstruction after oncological resection often leads to complications, which affect approximately 30% of patients. In recent decades, flap design has evolved towards perforator-based approaches to reduce this complication rate and reduce functional deficits. The aim of our systematic review and meta-analysis was to assess and compare the complication rate between perforator- and non-perforator-based reconstructions. Among 2576 screened studies, 49 met our inclusion criteria, encompassing 1840 patients. We found that the overall short-term surgical complication rate was comparable in patients receiving a perforator or a non-perforator flap, with a tendency towards fewer complications when using a perforator flap. Perforator flap-based reconstruction allows the surgeon to spare relevant anatomical structures and is therefore associated with fewer functional deficits, with a comparable complication rate.

**Abstract:**

Background: Patients with advanced vulvoperineal cancer require a multidisciplinary treatment approach to ensure oncological safety, timely recovery, and the highest possible quality of life (QoL). Reconstructions in this region often lead to complications, affecting approximately 30% of patients. Flap design has evolved towards perforator-based approaches to reduce functional deficits and (donor site) complications, since they allow for the preservation of relevant anatomical structures. Next to their greater surgical challenge in elevation, their superiority over non-perforator-based approaches is still debated. Methods: To compare outcomes between perforator and non-perforator flaps in female vulvoperineal reconstruction, we conducted a systematic review of English-language studies published after 1980, including randomized controlled trials, cohort studies, and case series. Data on demographics and surgical outcomes were extracted and classified using the Clavien–Dindo classification. We used a random-effects meta-analysis to derive a pooled estimate of complication frequency (%) in patients who received at least one perforator flap and in patients who received non-perforator flaps. Results: Among 2576 screened studies, 49 met our inclusion criteria, encompassing 1840 patients. The overall short-term surgical complication rate was comparable in patients receiving a perforator (*n* = 276) or a non-perforator flap (*n* = 1564) reconstruction (*p** > 0.05). There was a tendency towards fewer complications when using perforator flaps. The assessment of patients’ QoL was scarce. Conclusions: Vulvoperineal reconstruction using perforator flaps shows promising results compared with non-perforator flaps. There is a need for the assessment of its long-term outcomes and for a systematic evaluation of patient QoL to further demonstrate its benefit for affected patients.

## 1. Introduction

Vulvar cancer is rare among gynecological malignancies, accounting for less than 5% of gynecologic cancers. However, there has been a significant increase in incidence, especially in women under the age of 60 [1,2]. This imposes a serious physical and psychological burden on the affected women. The predominant subtype among vulvar carcinomas is squamous-cell carcinoma [3], which accounts for more than 90% of cases, followed by malignant melanoma, extramammary Paget’s disease, or, rarely, Bartolini gland carcinoma.

Over the past decades, the management of these neoplasms has significantly evolved. The advancement towards less radical surgical interventions has led to a noticeable reduction in complications, with a marked improvement in the quality of life (QoL) of patients [4,5]. A multidisciplinary approach integrating the expertise of gynecologic oncologists, radiologists, and plastic reconstructive surgeons is required to ensure optimal patient outcomes [5,6]. Surgeries such as wide local tumor excision, a standard procedure for squamous-cell carcinoma of the vulva, are performed up to the fascia. Partial or total vulvectomy and pelvic exenteration remain crucial for achieving local tumor control [7]. Depending on tumor size and the extent of the resected area, the resulting defect can be challenging to close. Traditionally, procedures without flap reconstruction have been associated with high complication rates, reaching nearly 60% [8]. The introduction of flap reconstructions, both perforator and non-perforator types, aimed to mitigate these adverse outcomes. However, despite these advancements, the complication rate remains concerningly high [9,10,11], with flap dehiscence, partial or complete flap loss, and infections at both the recipient and donor sites being the most prevalent issues.

Aiming to further decrease complications, there has been a paradigm shift from using muscle- or musculocutaneous flaps to fasciocutaneous perforator-based flaps (Figure 1) [12], which preserve the underlying muscle, thus preventing the loss of functionality at the donor site, and are believed to result in fewer complications [12]. Despite their theoretical advantages and successful application in other anatomical regions, evidence supporting their superiority in vulvoperineal reconstruction remains sparse [13,14,15].

Moreover, the role of radiotherapy introduces additional complexity to the surgical outcomes of vulvoperineal reconstruction. These patients are usually heavily radiated, and their proportion ranges from 27.9% in primary cancer with residual tumor to 65.1% in recurrent cases [16]. The acute and chronic adverse effects of radiotherapy significantly impact the risk of complications, affecting the choice and success of reconstructive strategies [17].

The aim of this systematic review and meta-analysis is to address the existing knowledge gap by providing a comprehensive evaluation of the current literature on the use of perforator and non-perforator flaps for vulvoperineal reconstruction after oncologic resection. To our knowledge, this is the first analysis that specifically compares the postoperative surgical outcomes of these two types of flap reconstructions. We hypothesize that the use of perforator flaps is associated with a lower risk of complications, particularly at the donor site, thereby potentially providing a significant improvement in postoperative recovery and quality of life for patients undergoing vulvoperineal reconstruction.

## 2. Materials and Methods

This systematic review and meta-analysis follows the Preferred Reporting Items for Systematic Reviews and Meta-Analyses (PRISMA) recommendations. It was registered in the PROSPERO International prospective register of systematic reviews. The registration took place on 1 March 2023 (ID CRD42023403496).

### 2.1. Search Strategy

We conducted a comprehensive systematic literature search to identify articles addressing pelvic cancers and reconstructive procedures for vulvoperineal defects, with a specific focus on the utilization of flap techniques. The search strategy was developed by a medical information specialist. The search syntax was composed and optimized in Embase (Elsevier, Amsterdam, The Netherlands), from where it was translated for other databases using publicly available macros [18]. The bibliographic databases Embase (Elsevier), Medline (Ovid, New York, NY, USA), and Web of Science Core Collection (Clarivate, Philadelphia, PA, USA) were searched using database-specific subject headings and text words (last search 28 September 2022). No language or publication date restrictions were applied but conference abstracts were excluded from the search. The full search strategies can be found in the electronic Appendix A.

### 2.2. Inclusion and Exclusion Criteria

The selection process focused on all articles considering either perforator or non-perforator flaps for vulvoperineal reconstruction after oncologic resection. To enhance clarity, we organized flaps into two distinct groups: perforator flaps, which encompass options like the pedicled profunda artery perforator (PAP), anterolateral thigh (ALT), deep inferior epigastric perforator (DIEP), superior gluteal (SGAP), and inferior gluteal (IGAP) flap, and non-perforator flaps, including muscle-based techniques such as the vertical rectus abdominis myocutaneous (VRAM) flap, gracilis flap, and tensor fasciae latae flap as well as local random-pattern fasciocutaneous flaps like the lotus petal flap, V-Y fasciocutaneous flaps, and the gluteal fold flap, among others.

The selected articles were expected to provide information on postoperative outcomes for each intervention group. These postoperative outcomes included various aspects, such as the overall complication rate and complications that occurred at both the donor and recipient sites. These complications were well defined and included perineal infections, flap necrosis, partial or complete flap loss, dehiscence rate, and infection rate. Studies were eligible if they reported one or more of these predefined outcomes. The inclusion and exclusion criteria are summarized in Table 1. Specifically, we excluded case reports and small case series with fewer than seven patients. In addition, reviews, commentaries, letters to the editor, cadaver studies, animal studies, and articles not written in English were excluded from our analysis.

### 2.3. Selection Process and Data Extraction

First, three authors performed a calibration phase with 100 abstracts to ensure that they uniformly applied the predefined inclusion and exclusion criteria. Subsequently, titles and abstracts were independently screened by two authors to identify all potentially relevant papers. The selected studies were then reviewed in full text by two independent authors. If the inclusion criteria were met, the data were independently extracted by both authors into a standardized Excel file. Any disagreements in the selection process and data extraction were resolved in discussion with the senior author. The following data were extracted: study design; country and timing of study; patient demographics, such as age, BMI, their oncologic diagnosis, details of the reconstruction procedures performed, and surgical outcomes, along with screening for complications at donor and recipient site; and assessments of patient quality of life.

### 2.4. Clavien–Dindo Classification

In order to ensure consistency in the assessment of complications across various studies, we standardized the categorization of complications using the Clavien–Dindo classification [19].

### 2.5. Statistics

To determine whether the use of perforator flaps is associated with a risk of complication, we used a random-effects meta-analysis to derive a pooled estimate of the frequency of complications as a percentage in patients who received at least one perforator flap and in patients who received non-perforator flaps. Pooled complication frequency was based on reported complication proportions and 95% Wilson confidence intervals. Studies reporting on either treatment were included as two one-arm studies in this analysis. We used the empirical Bayesian approach for estimation and assessed between-study variance (τ^2^), and heterogeneity as Cochran’s Q test, H^2^, and I^2^. We calculated a pooled effect size, e.g., risk difference, using meta-regression with treatment as the only independent variable and visualized the results as a forest plot. Two studies [20,21] reported complications only at the flap level, not at the patient level. For comparability, we approximated the proportion of patients with complications assuming each patient had at most one flap complication. 

We performed three sensitivity analyses to assess whether the surgical procedure could explain the difference in complication rates. First, we included further variables as covariates in the meta-regression; “recent study” was defined as year of publication ≥ 2015 and “large study” was defined as sample size ≥50. Second, we grouped patients who received flaps of both types as “non-perforator flap” and repeated the main analysis based on this grouping. Finally, we restricted the analysis to studies which reported on either treatment and hence provided a risk difference and calculated a pooled effect size using a random-effects meta-analysis with empirical Bayesian estimation. We refrained from meta-analyzing quality-of-life data as only a few studies reported these outcomes.

All analyses were conducted using Stata 16.0 (StataCorp LLC, College Station, TX, USA).

## 3. Results

The initial search query resulted in 4710 records; after removing 2134 duplicates, 2576 records remained. After initial title and abstract screening, 395 articles were assessed in full text. Forty-nine studies fulfilled the selection criteria. Among them, thirty-two studies analyzed non-perforator flaps [9,10,20,22,23,24,25,26,27,28,29,30,31,32,33,34,35,36,37,38,39,40,41,42,43,44,45,46,47,48,49,50], only nine studies focused on perforator flaps [21,51,52,53,54,55,56,57,58], and eight studies focused on both [59,60,61,62,63,64,65,66] (Figure 2).

### 3.1. Study Characteristics

The 49 studies were conducted in Europe (*n* = 29) [9,10,20,22,24,25,26,27,28,30,31,34,35,36,38,39,40,41,42,43,45,47,50,52,54,57,58,62,65], the United States (*n* = 5) [29,44,46,48,49], and Asia (*n* = 15) [21,23,32,33,37,51,53,55,56,59,60,61,63,64,66] and were published between 1990 and 2022. Two studies were multicentric [28,50]; the others were monocentric. One study [46] was prospective and all others were retrospective. A total of 1840 female patients were included for analysis in our study, with 2077 (84%) non-perforator flaps and 384 (16%) perforator flaps (Table 2).

### 3.2. Overall Complications

In this random-effects meta-analysis, the incidence of complications was lower in patients who underwent a perforator flap reconstruction, although this difference did not reach statistical significance. Specifically, the explained variance in the outcome (R^2^) was 0%, indicating that perforator flaps did not have an impact on the frequency of complications (**p* > 0.05). Heterogeneity among the studies (I^2^) was as large as 91%; hence, 91% of the observed variation among studies is due to the heterogeneity of findings rather than chance. Between-study variance τ^2^ was 0.05; thus, it was larger than the estimated treatment effect as absolute. Cochran’s Q test showed that study results differed, as the studies reported significantly different frequencies of complications (***p* < 0.001) (Table 3, Figure 3).

### 3.3. Complications According to the Clavien–Dindo Classification

The overall complications graded according to the Clavien–Dindo classification are shown in Figure 4. Most of the reported complications were minor (Clavien–Dindo I) and were observed with a frequency of occurrence between 0 and 90% in the studies. The frequency of the second most commonly reported complication (Clavien–Dindo III) was observed in between 0 and 69% of cases. We did not find any complications with Clavien–Dindo grade IV or higher. Donor site complications are shown in Figure 5. Several studies did not report on their donor site complications (“Not reported” in Figure 5). Nine studies in the non-perforator flap [9,23,32,36,39,40,48,49,60] and three studies in the perforator flap group [52,56,59] reported on complications at the donor site. The frequency of donor site complications ranged from 0 to 10% in the included studies.

Recipient site complications are shown in Figure 6. The distribution of complications on the recipient site according to the Clavien–Dindo classification is comparable to the overall complications. 

### 3.4. Sensitivity Analysis

First, adjustment for year of publication and study size did not show an association with the frequency of complications and did not alter the effect estimate of treatment, model fit, and heterogeneity parameters. Secondly, patients who received both a perforator and a non-perforator flap were grouped as non-perforator flap, which yielded similar results as the main analysis (Table 4, Figure 7). 

Thirdly, when we limited the analysis to studies that specifically addressed both perforator and non-perforator flap reconstructions, we were left with six studies only. This exclusion resulted in the omission of a substantial number of studies, precisely 43 out of the total 49 studies initially considered. The pooled risk difference suggested a potential advantage of perforator flap reconstruction; however, the reported risk difference was different (*p* of Cochran’s Q test was <0.001) and the heterogeneity was very high (I^2^ was 92%) (Figure 8).

As a result, a meta-regression of these six studies was neither feasible nor sensible, as the pooled estimate and heterogeneity measures from the meta-analysis above correspond exactly to the results derived from a constant-only meta-regression model. The criterion “size of study ≥ 50 patients” would yield just one group, and “publication after 2014” would comprise one single study only. Hence, adjustments as performed in the main analysis would not be appropriate.

### 3.5. Satisfaction and Quality of Life

Patient satisfaction was analyzed in eight studies [21,25,28,32,51,53,58,60] including only 177 patients (9.6%). These eight studies showed a wide range in the proportion of satisfied patients, ranging from 38% to 100% (Figure 9). The low percentage of satisfied patients in the study by Shin et al. [51] and Chang et al. [21] is explained by the fact that only a small proportion of patients reported their level of satisfaction to the study team. Five studies [25,28,44,49,60] from the non-perforator group addressed returning to sexual activity after surgery (Figure 10). Tan et al. [60] also used one perforator flap but did not report on the sexual activity of this patient specifically. Although not reported in the analyzed studies, the effect of adjunctive radiotherapy may lead to radiation dermatitis, with a long-term negative impact on QoL [17].

## 4. Discussion

With the continuous development of well-established systemic treatments, such as immunotherapy, and with the improvement in radiation therapy as well as development of surgical treatment options over the past decades, there is a significant increase in cancer survivors wishing for good QoL. Novel and innovative treatment approaches in the field of plastic surgery often encounter obstacles in their acceptance (process) and take a long time to become established across partner disciplines. It is therefore crucial for reconstructive surgeons to show collaborating specialties that exceptional results can be achieved with minimal complications. We must also constantly focus on innovation and improvement in our work. Thirty years ago, in 1994, arguably the first autologous breast reconstruction was performed with a DIEP flap, which at that time had already demonstrated its advantages over conventional methods, with lower donor site morbidity [67]. Nowadays, perforator flaps like the DIEP flap have been established as the gold standard in autologous breast reconstruction and are integral components in the armamentarium of high-volume medical centers. This illustrates the inertia in implementing innovations in existing systems. 

The innovation of perforator flaps in general and the convincing results from other anatomical regions, coupled with the improvement in patients’ QoL, drove us to investigate the role of perforator flaps in the reconstruction of vulvoperineal defects. We therefore believe that this systematic review is, to the best of our knowledge, not only the first, but also the most comprehensive analysis of surgical outcomes comparing perforator and non-perforator flaps in female patients with vulvoperineal defects. Of the 2576 studies identified by the literature search, only 49 studies could ultimately be included in this review. Most of these studies assessed the outcomes of non-perforator flaps, with only eight studies specifically examining the results of perforator flaps. This shows that the literature on reconstructive procedures with perforator flaps in the vulvoperineal region is sparse.

### 4.1. Surgical Complications

There is a large range of surgical options available for closing vulvoperineal defects. These include primary closure, locoregional random-patterned, axially patterned, and perforator-based flaps, as well as musculocutaneous or perforator flaps. 

The introduction of flap reconstruction as such has already been shown to reduce the complication rate significantly [68]. However, the high complication rate remains a challenge because the available reconstructive procedures are still associated with high morbidity at the recipient and donor sites. A major advantage of perforator flaps is their design, consisting only of the skin and subcutaneous fat tissue, leaving the underlying muscles and nerves intact [69]. Harvesting perforator flaps may keep other surgical options open, compared to random-pattern flaps, and they show a stable vascular territory compared to axial pattern flaps. 

Consequently, there is usually less morbidity at the donor site, as muscle function is maintained. In many of the included studies (49%), there is unfortunately no clear distinction made between complications at the donor vs. the recipient site. On the other hand, research has demonstrated that muscle-based reconstructions have a high rate of donor site morbidity [70,71]. We therefore presume that the proportion of donor site complications in the included studies is under-reported. 

For an effective comparison of the surgical outcomes of different studies, it is crucial that the complications are reported in a standardized manner. One approach commonly employed for this purpose is the Clavien–Dindo classification, which assesses the severity of complications according to the treatment they require [19]. In the studies we included, only a minority of them reported surgical outcomes using this classification method, which makes the comparison of surgical complications challenging.

### 4.2. Potential Importance of Perforator Flaps in Female Vulvoperineal Reconstruction

Our results potentially suggest that the use of perforator flaps in vulvoperineal reconstruction is associated with fewer complications compared to conventional flap reconstruction. The fact that this is a trend only may be attributable to the lack of studies directly comparing perforator versus non-perforator flaps and the high heterogeneity of the studies, which may lead to a loss of predictive power. It is interesting to note, however, that perforator flaps have become established as one of the leading procedures in breast reconstruction and show high success rates with a relatively low complication rate [72,73,74]. The same observation has been made in head and neck reconstructive surgery, with convincing results pertaining to the use of perforator flaps [75,76]. 

### 4.3. Assessment of Patient Satisfaction and Quality of Life

Only eight of forty-nine studies examined patient satisfaction and QoL. The finding that QoL continues to be under-reported and therefore is not the focus of outcomes is consistent with the current literature in the field of pelvic reconstruction. Witte et al. [77] compared the various options for flap reconstruction after pelvic exenteration, one of their secondary outcomes being the assessment of QoL. QoL was recorded by <10% of the included studies, again consistent with our findings. The vulvoperineal region plays an essential role in women’s QoL because it is responsible for vital functions such as micturition, defecation, reproduction, and psychosexual integrity. The vulvoperineal area is essential for the functioning of basic needs, such as intestinal voiding, urination, and sexual intercourse. Sexual function is an especially important factor of psychological well-being. Therefore, the assessment of QoL as an outcome parameter in this anatomical region should not be underestimated.

### 4.4. Limitations

Many of the included studies lack precise descriptions of complications and their management, which can result in both the over- and underestimation of complication rates, thereby complicating the Clavien–Dindo classification. To limit these biases, we extracted complications as they were listed. In cases where the management of complications was not provided, we made assumptions based on best practices and subsequently categorized them accordingly. Some studies did not explicitly report donor site complications. In these cases, we assumed that the complications occurred at the recipient site, as complications are more frequent there. This assumption may have led to an overestimation of the complication rates at the recipient site of the studies concerned. Furthermore, the included studies showed a notably high degree of heterogeneity, which can be explained by the small sample sizes, the population’s complexity, and the intervention being studied. 

A major limitation of this study is that the impact of radiation was not analyzed, mainly due to lack in reporting in the chosen studies. In the included literature, the effect of adjuvant or neoadjuvant radiation is not extensively addressed, despite playing an important part in outcomes and QoL.

Finally, the small proportion of studies that compared perforator and non-perforator flaps in a direct manner limits the conclusiveness of our sensitivity analysis. 

### 4.5. Perforator Flaps as a Valid Choice in the Reconstructive Armamentarium

Perforator flaps have become established as a safe method with a good functional and aesthetic outcomes in multiple areas of reconstructive surgery. Perforator flaps can be chosen based on the reconstruction demands. The ALT flap provides stability due to its thick fascia. The PAP flap, with its thin dermis, is more pliable, if needed. Furthermore, sensitized flaps (ALT and PAP) can be performed.

It is noteworthy that the complication rates between perforator and non-perforator flaps are comparable, with a promising trend indicating fewer complications with perforator flaps. However, these results should be interpreted with the reservation of a possible bias due to the large heterogeneity between the analyzed studies. With careful consideration of the patient’s individual circumstances, wishes, and expectations and the surgeon’s level of experience, perforator flaps should be considered a viable option for the treatment of vulvoperineal defects.

## 5. Conclusions and Future Direction

Vulvoperineal reconstruction using perforator flaps shows promising results with a trend towards lower complication rates compared with conventional procedures. While technically challenging, this approach offers the potential to preserve important anatomical structures and, consequently, maintain functionality. To demonstrate the efficacy of perforator flaps in vulvoperineal reconstruction as successfully as in the field of breast and head and neck reconstruction and to validate or challenge the trend we have presented, it is essential to further investigate this in more extensive, direct comparative studies between perforator and non-perforator flaps in vulvoperineal reconstruction. Moreover, it is critical to comprehensively assess patient QoL through the systematic implementation and assessment of patient-reported outcome measures in clinical practice and research.

## Figures and Tables

**Figure 1 cancers-16-02213-f001:**
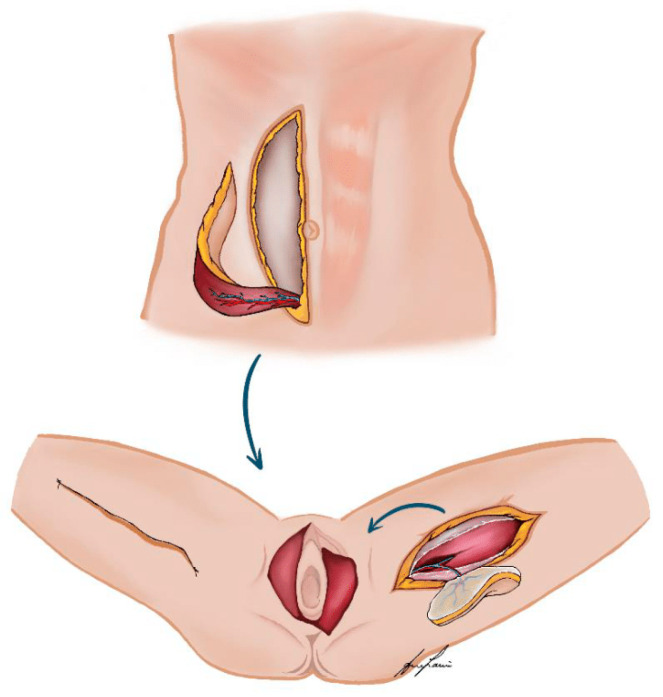
Illustrated examples of muscle- or musculocutaneous flaps (VRAM—flap from the abdominal wall) and fasciocutaneous perforator-based flaps (PAP—flap from the left thigh, donor site closure as vertical thigh lift) as local vulvoperineal reconstruction options.

**Figure 2 cancers-16-02213-f002:**
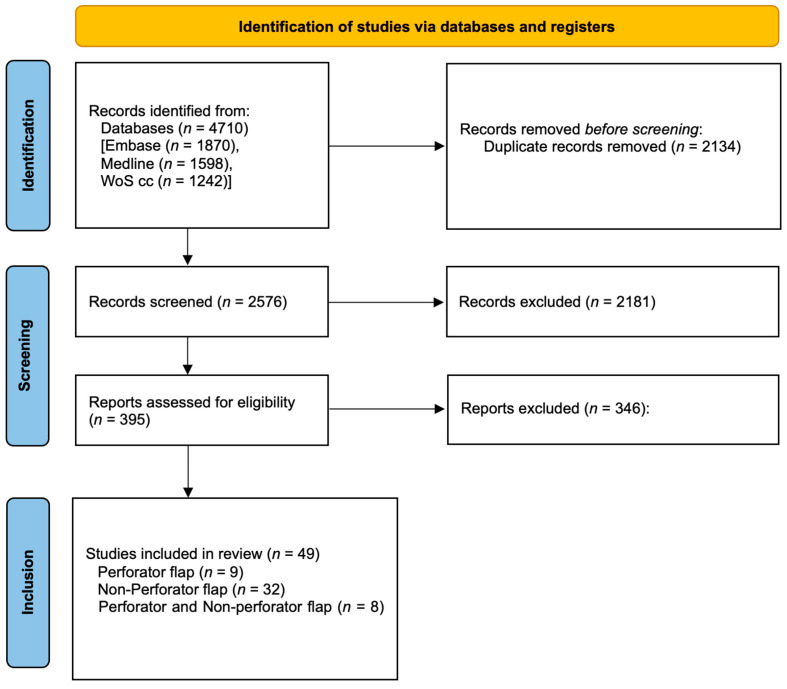
PRISMA flow chart.

**Figure 3 cancers-16-02213-f003:**
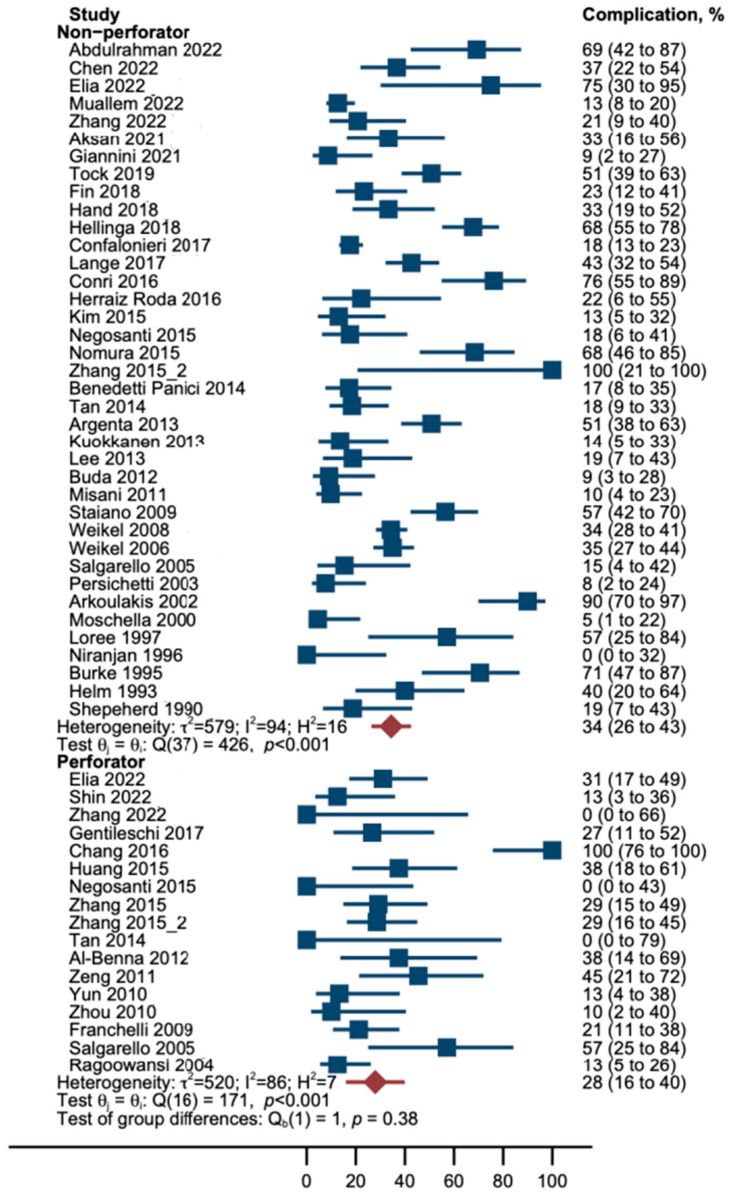
Forest plot; Proportion of patients who suffered from complications (blue boxes) by treatment. Note that confidence intervals of proportions (horizontal lines) are asymmetrical because we used Wilson’s method for calculation of interval boundaries to account for small samples and few events. Red diamonds and lines represent pooled proportions with Wilson CI [9,10,20,21,22,23,24,25,26,27,28,29,30,31,32,33,34,35,36,37,38,39,40,41,42,43,44,45,46,47,48,49,50,51,52,53,54,55,56,57,58,59,60,61,62,63,64,65,66].

**Figure 4 cancers-16-02213-f004:**
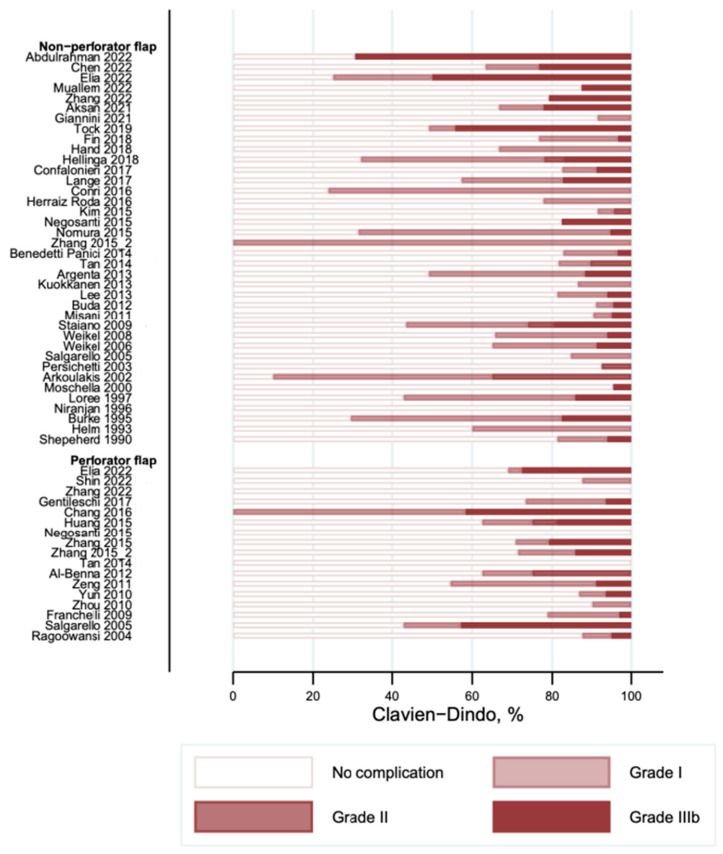
Overall complications according to Clavien–Dindo classification [9,10,20,21,22,23,24,25,26,27,28,29,30,31,32,33,34,35,36,37,38,39,40,41,42,43,44,45,46,47,48,49,50,51,52,53,54,55,56,57,58,59,60,61,62,63,64,65,66].

**Figure 5 cancers-16-02213-f005:**
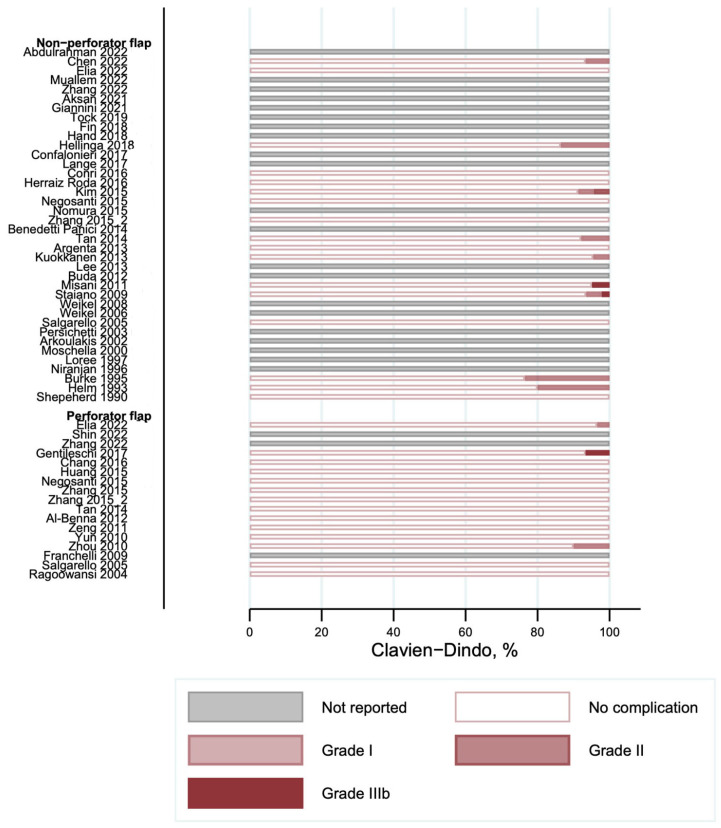
Donor site complications according to Clavien–Dindo classification [9,10,20,21,22,23,24,25,26,27,28,29,30,31,32,33,34,35,36,37,38,39,40,41,42,43,44,45,46,47,48,49,50,51,52,53,54,55,56,57,58,59,60,61,62,63,64,65,66].

**Figure 6 cancers-16-02213-f006:**
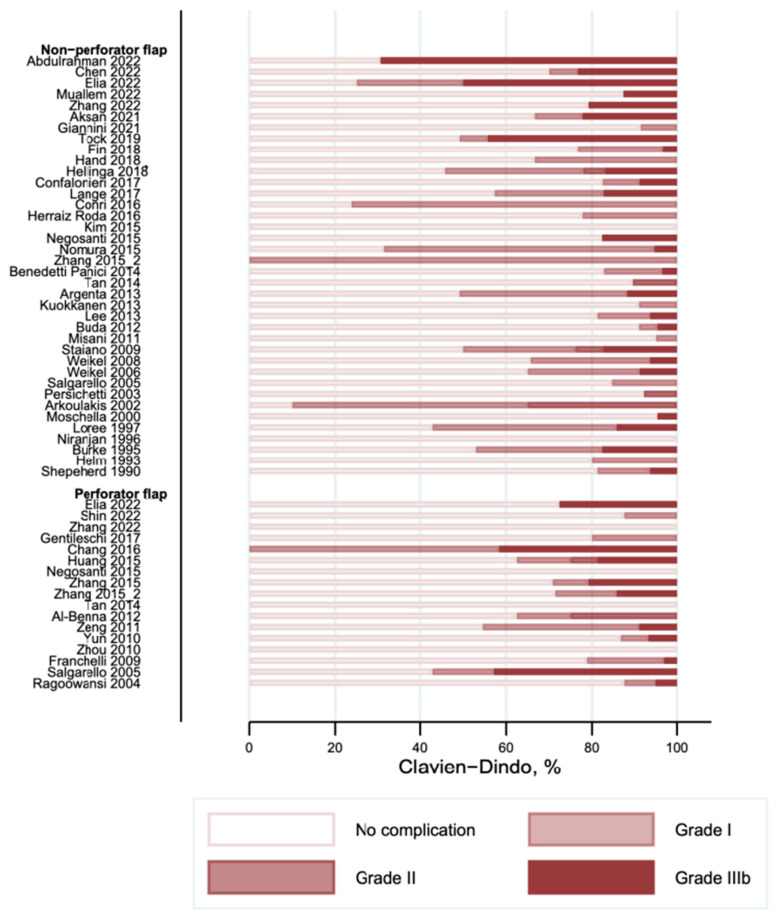
Recipient site complications according to Clavien–Dindo classification [9,10,20,21,22,23,24,25,26,27,28,29,30,31,32,33,34,35,36,37,38,39,40,41,42,43,44,45,46,47,48,49,50,51,52,53,54,55,56,57,58,59,60,61,62,63,64,65,66].

**Figure 7 cancers-16-02213-f007:**
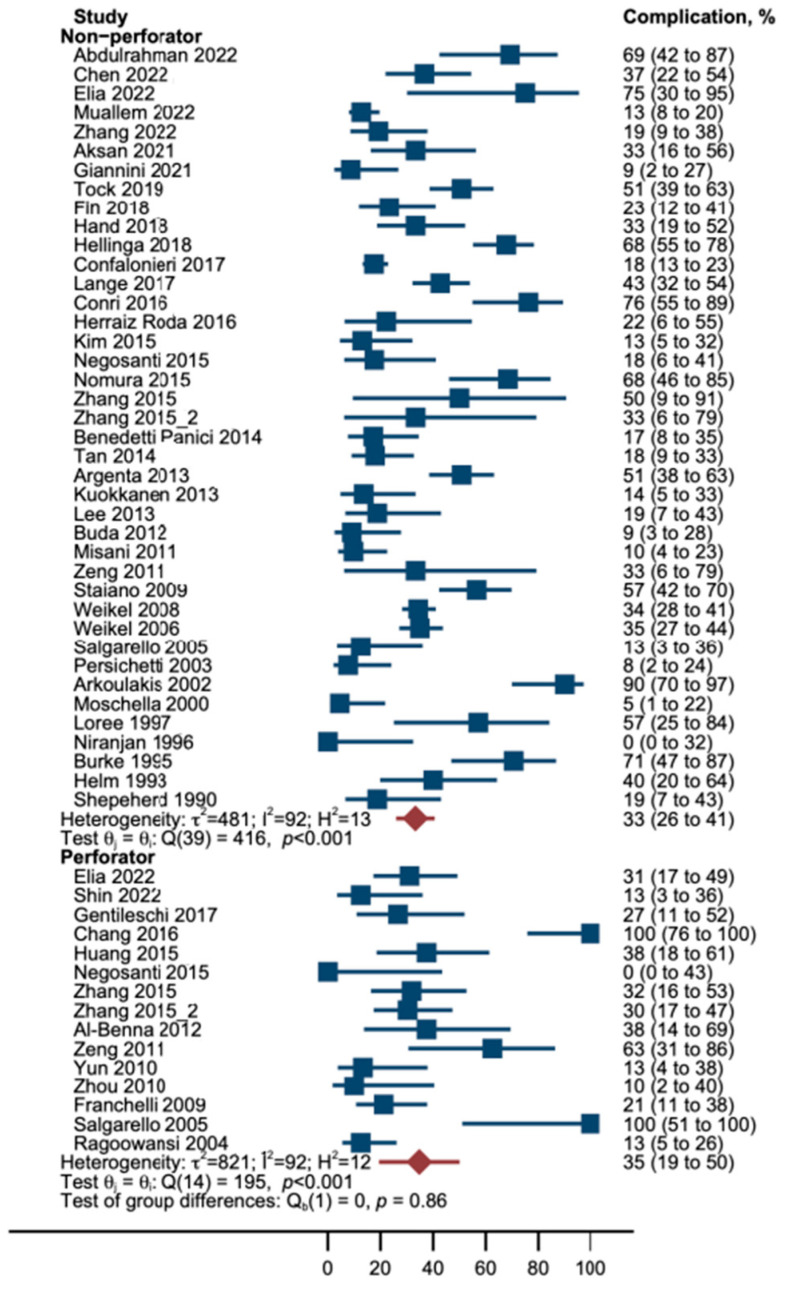
Forest plot of sensitivity analysis 2: Association of treatment and proportion (blue boxes) of complications. Note that confidence intervals of proportions (horizontal lines) are asymmetrical because we used Wilson’s method for calculation of interval boundaries to account for small samples and few events. Red diamonds and lines represent pooled proportions with Wilson CI [9,10,20,21,22,23,24,25,26,27,28,29,30,31,32,33,34,35,36,37,38,39,40,41,42,43,44,45,46,47,48,49,50,51,52,53,54,55,56,57,58,59,60,61,62,63,64,65,66].

**Figure 8 cancers-16-02213-f008:**
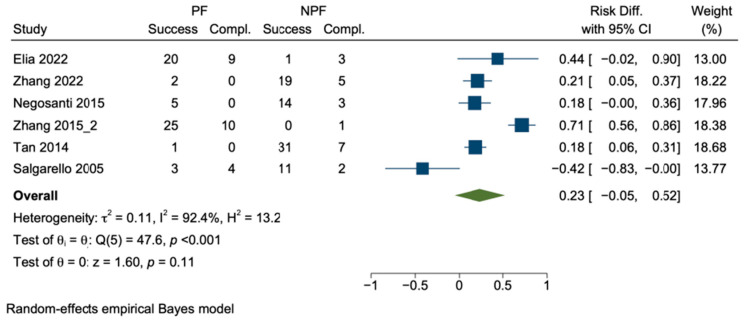
Sensitivity analysis 3: Pooled risk difference. Boxes represent differences in the risk of complication after perforator vs. non-perforator flap reconstruction with confidence interval, e.g., effect sizes. Green diamond shows pooled risk difference. PF perforator flap; NPF non–perforator flap [59,60,61,62,64,65].

**Figure 9 cancers-16-02213-f009:**
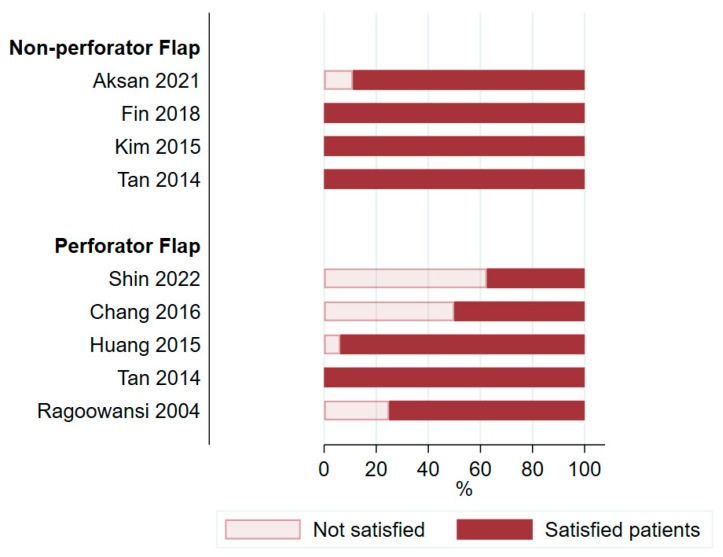
Patient satisfaction after surgery [21,25,28,32,51,53,58,60].

**Figure 10 cancers-16-02213-f010:**
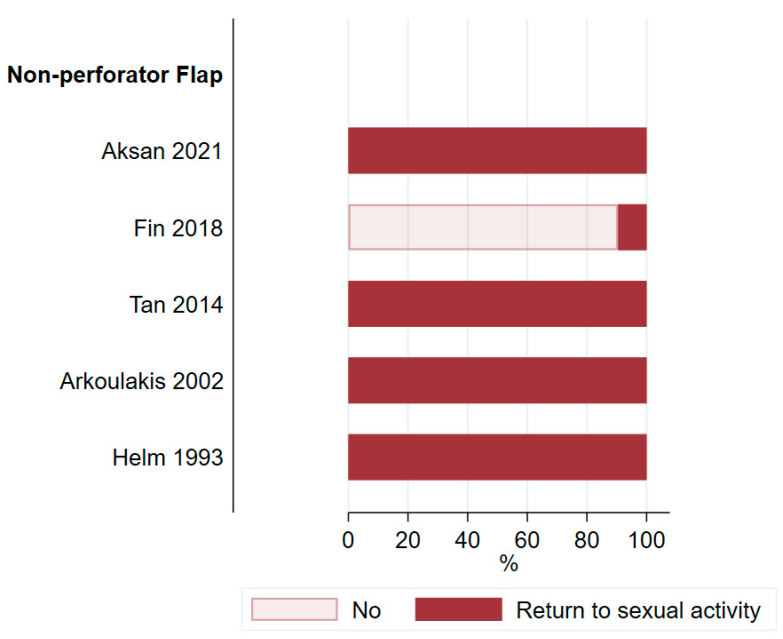
Proportion of patients returning to sexual activity after surgery [25,28,44,49,60].

**Table 1 cancers-16-02213-t001:** Inclusion and exclusion criteria.

PICOS	Inclusion	Exclusion
Populations	Female adults with vulvoperineal reconstruction with perforator or non-perforator flap	Cadaveric, animal studies
Intervention	Perforator or non-perforator flap for vulvoperineal reconstruction following oncologic resection	Other reconstruction techniques like primary closure or net implementation
Comparator	The study analysis compared postoperative surgical outcome parameters	
Outcomes	Main outcome: complications like infection rate, dehiscence, partial or total flap necrosis graded according to Clavien–Dindo classification	Studies that do not report main outcome
Study design	Randomized controlled trials, comparative studies, and case series ≥7 patients	Reviews, meta-analyses, case reports, unpublished studies, and non-English language studies

**Table 2 cancers-16-02213-t002:** Study overview. # number.

Study	# Flaps	Non-Perforator	Perforator
Abdulrahman 2022 [22]	18	Gracilis myocutaneous flap (7), IGAM (1), local fasciocutaneous flap (2), VRAM (8)	
Chen 2022 [23]	30	VRAM (30)	
Elia 2022 [59]	55	Gracilis myocutaneous flap (3), local fasciocutaneous flap (2), VRAM (1)	ALT (3), local perforator fasciocutaneous flap (23), PAP (23)
Muallem 2022 [24]	126	Flap combination (22), local fasciocutaneous flap (84), myocutaneous flap (20)	
Shin 2022 [51]	27		Local perforator fasciocutaneous flap (27)
Zhang 2022 [61]	34	Local fasciocutaneous flap (30), VRAM (2)	ALT (2)
Aksan 2021 [25]	31	Local fasciocutaneous flap (29), skin graft (1), VRAM (1)	
Giannini 2021 [26]	40	Local fasciocutaneous flap (40)	
Tock 2019 [27]	61	Gluteal thigh flap (16), local fasciocutaneous flap (45)	
Fin 2018 [28]	59	Local fasciocutaneous flap (59)	
Hand 2018 [29]	42	Local fasciocutaneous flap (42)	
Hellinga 2018 [9]	89	Local fasciocutaneous flap (89)	
Confalonieri 2017 [10]	365	Local fasciocutaneous flap (365)	
Gentileschi 2017 [52]	16		ALT (16)
Lange 2017 [20]	114	Local fasciocutaneous flap (114)	
Chang 2016 [21]	19		PAP (19)
Conri 2016 [30]	36	Local fasciocutaneous flap (36)	
Herraiz Roda 2016 [31]	17	Local fasciocutaneous flap (17)	
Huang 2015 [53]	27		Local perforator fasciocutaneous flap (16), PAP (11)
Kim 2015 [32]	41	Local fasciocutaneous flap (41)	
Negosanti 2015 [62]	33	Local fasciocutaneous flap (28)	DIEP (5)
Nomura 2015 [33]	19	Local fasciocutaneous flap (16), non-specified flap (3)	
Zhang 2015 [63]	27	Gracilis myocutaneous flap (1), TRAM (1)	ALT (24), DIEP (1)
Zhang 2015_2 [64]	40	Gracilis myocutaneous flap (2), TRAM (1)	ALT (24), DIEP (6), pudendal thigh fasciocutaneous flap (7)
Benedetti Panici 2014 [34]	29	Local fasciocutaneous flap (29)	
Tan 2014 [60]	72	Gracilis myocutaneous flap (36), local fasciocutaneous flap (21), skin graft (11), VRAM (3)	ALT (1)
Argenta 2013 [35]	80	Local fasciocutaneous flap (80)	
Kuokkanen 2013 [36]	22	Local fasciocutaneous flap (22)	
Lee 2013 [37]	27	Local fasciocutaneous flap (27)	
Al-Benna 2012 [54]	13		Local perforator fasciocutaneous flap (13)
Buda 2012 [38]	38	Local fasciocutaneous flap (38)	
Misani 2011 [39]	69	Local fasciocutaneous flap (69)	
Zeng 2011 [66]	14	Local fasciocutaneous flap (3)	ALT (11)
Yun 2010 [55]	24		PAP (24)
Zhou 2010 [56]	10		ALT (10)
Franchelli 2009 [57]	53		Local perforator fasciocutaneous flap (53)
Staiano 2009 [40]	53	Gracilis myocutaneous flap (4), latissimus dorsi muscle flap (1), local fasciocutaneous flap (26), tensor fasciae latae flap (1), VRAM (21)	
Weikel 2008 [41]	207	Gluteal thigh flap (54), gracilis myocutaneous flap (5), local fasciocutaneous flap (123), tensor fasciae latae flap (9), VRAM (16)	
Weikel 2006 [42]	123	Gluteal thigh flap (40), gracilis myocutaneous flap (4), local fasciocutaneous flap (58), tensor fasciae latae flap (8), VRAM (13)	
Salgarello 2005 [65]	31	Local fasciocutaneous flap (18), VRAM (4)	Pudendal thigh fasciocutaneous flap (9)
Ragoowansi 2004 [58]	56		Local perforator fasciocutaneous flap (56)
Persichetti 2003 [43]	26	Local fasciocutaneous flap (26)	
Arkoulakis 2002 [44]	36	Local fasciocutaneous flap (36)	
Moschella 2000 [45]	22	Local fasciocutaneous flap (22)	
Loree 1997 [46]	13	Gracilis (2), local fasciocutaneous flap (10), VRAM (1)	
Niranjan 1996 [47]	13	Local fasciocutaneous flap (13)	
Burke 1995 [48]	18	Gracilis myocutaneous flap (18)	
Helm 1993 [49]	30	Local fasciocutaneous flap (30)	
Shepeherd 1990 [50]	16	VRAM (16)	

**Table 3 cancers-16-02213-t003:** Meta-regression: Association of treatment and proportion of complications.

Variable	Association with Outcome		Model Fit		Heterogeneity among Studies			
	Coefficient (95% CI)	*p*	R^2^	*p* *	I^2^ (%)	H^2^	τ^2^	*p* **
Treatment	−0.06 (−0.20 to 0.09)	0.459	0	0.450	91	11	0.05	<0.001
Treatment	−0.06 (−0.21 to 0.08)	0.413	1	0.299	90	10	0.05	<0.001
Year ≥ 2015	0.09 (−0.04 to 0.22)	0.175						
Treatment	−0.04 (−0.20 to 0.11)	0.599	0	0.617	90	10	0.05	<0.001
# Patients ≥ 50	0.06 (−0.13 to 0.25)	0.540						
Treatment	−0.05 (−0.21 to 0.10)	0.522	0	0.457	90	10	0.05	<0.001
Year ≥ 2015	0.09 (−0.05 to 0.22)	0.203						
# Patients ≥ 50	0.04 (−0.14 to 0.23)	0.654						

* *p* of regression model testing whether explanatory variables are associated with dependent variable. ** *p* of Cochran’s Q test of residual homogeneity testing the H_0_ that all study results are equal. # number

**Table 4 cancers-16-02213-t004:** Sensitivity analysis 2: association of treatment and proportion of complications.

	Association with Outcome		Model Fit		Heterogeneity among Studies			
Variable	Coefficient (95% CI)	*p*	R^2^	**p*	I^2^ (%)	H^2^	τ^2^	***p*
Treatment	0.01 (−0.14 to 0.17)	0.880	0	0.871	91	11	0.05	<0.001
Treatment	0.01 (−0.14 to 0.16)	0.917	0	0.561	91	11	0.05	<0.001
Year ≥ 2015	0.06 (−0.07 to 0.20)	0.368						
Treatment	0.03 (−0.13 to 0.19)	0.737	0	0.753	91	11	0.05	<0.001
# Patients ≥ 50	0.07 (−0.12 to 0.26)	0.472						
Treatment	0.02 (−0.14 to 0.18)	0.792	0	0.745	91	10	0.06	<0.001
Year ≥ 2015	0.06 (−0.08 to 0.19)	0.421						
# Patients ≥ 50	0.06 (−0.13 to 0.25)	0.545						

* *p* of regression model testing whether explanatory variables are associated with dependent variable. ** *p* of Cochran’s Q test of residual homogeneity testing the H_0_ that all study results are equal. # number.

## Data Availability

Data are contained within the article and the Appendix A.

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
