# Peer review of "Perforator versus Non-Perforator Flap-Based Vulvoperineal Reconstruction—A Systematic Review and Meta-Analysis"

_cancers, 2024, doi:10.3390/cancers16122213_

Round 1

Reviewer 1 Report

Comments and Suggestions for Authors

The introduction contains relevant background information but could benefit from being more focused.

While the literature review is thorough, it feels somewhat disorganized. The flow could be improved to better connect the advancements in therapeutic management with the specific focus on flap reconstruction.

It's not clear how these macros ensure the accuracy and completeness of the search. A brief explanation of this process or a justification for using these macros would strengthen the validity of the search strategy.

The data extraction process is thorough, but the methods for handling missing data or inconsistent reporting across studies are not discussed.

The description of statistical findings is detailed, but the significance of "R2 = 0%" and "large heterogeneity (I2 > 75%, H2 > 1)" needs further interpretation regarding its impact on the study conclusions.

For donor site complications, stating that several studies did not report them and labeling this "Not Applicable" in Figure 5 without further explanation is insufficient.

The discussion on sexual activity post-surgery is brief and should address why this is a critical outcome and how its underreporting affects the study's findings.

The comparison of perforator flaps in other surgical fields (breast, head, and neck reconstruction) is useful. However, the assertion that perforator flaps result in fewer complications should be tempered with more detailed statistical evidence from the study

Author Response

Dear Reviewer 1,

Thank you for your time and efforts considering our paper. 

Please find our answers for your comments and questions in our Answer-Sheet.

Kind regards,

Dr. med. L. Kouba and S. Wendelspiess

with Prof. E. Kappos and PD. Dr. I. Tarek

Reviewer 2 Report

Comments and Suggestions for Authors

I give you the following comment to enhance the readability and understanding of your manuscript.

    How common are difficulties in patients having repair of the vulvoperineal defect following oncological resection, and how has flap design changed to meet these problems? Which particular kind of flap design aims to lessen problems and functional deficits? How many papers were examined in the meta-analysis and systematic review to determine the frequency of complications associated with various flap designs? What were the overall conclusions on the rates of short-term complications following flaps with or without perforators?

     What are the main objectives of a comprehensive treatment strategy for patients with advanced vulvoperineal cancer, and why is one necessary?What is the estimated proportion of patients in this area that have problems after reconstructions? When compared to non-perforator-based techniques, how can flap designs based on perforators aid in the preservation of anatomical structures? What, notwithstanding the surgical challenge of elevating the perforator-based techniques, is still up for discussion regarding their superiority?

    Which papers comparing perforator and non-perforator flaps in female vulvoperineal reconstruction were chosen for the systematic review based on what criteria? In what ways were data retrieved and surgical outcomes categorised for this review? Which technique was applied to get a pooled estimate of the frequency of complications for each of the two flap types? How many patients received each kind of flap in the studies that made up the review's sample?

     Regarding the overall surgical short-term complication rate comparing perforator and non-perforator flaps, what did the meta-analysis reveal? Was there a hint as to which kind of flap might cause fewer problems? How frequently was the quality of life of the patients evaluated in the included studies? What further information, according to the authors, is required to more clearly illustrate the advantages of perforator flaps for afflicted patients?

    Based on this analysis, what conclusions can be made regarding the usage of perforator flaps versus non-perforator flaps in vulvoperineal reconstruction? Why is it necessary to evaluate quality of life systematically and measure long-term outcomes? How many patients and study records were examined to arrive at these findings? Even if the rates of short-term complications are similar, what are the possible advantages of utilising perforator flaps? 

minor:

1. Please increase the size of Figure 3 to Figure 4.

2. It would be appreciated if you could check the references and grammar errors in your manuscript.

3. Please add it to the conclusion section with "conclusion and future direction.".

Author Response

Dear Reviewer 2,

Thank you for your time and efforts considering our paper. 

Please find our answers for your comments and questions in our Answer-Sheet.

Kind regards,

Dr. med. L. Kouba and S. Wendelspiess

with Prof. E. Kappos and PD. Dr. I. Tarek

Round 2

Reviewer 1 Report

Comments and Suggestions for Authors The manuscript has been sufficiently improved to warrant publication in Cancers